# Influencing Factors on the Oncuria™ Urinalysis Assay: An Experimental Model

**DOI:** 10.3390/diagnostics11061023

**Published:** 2021-06-03

**Authors:** Kaoru Murakami, Ian Pagano, Runpu Chen, Yijun Sun, Steve Goodison, Charles J. Rosser, Hideki Furuya

**Affiliations:** 1Samuel Oschin Comprehensive Cancer Institute, Cedars-Sinai Medical Center, Los Angeles, CA 90048, USA; kaoru.murakami@cshs.org (K.M.); charles.rosser@cshs.org (C.J.R.); 2Cancer Prevention and Control Program, University of Hawaii Cancer Center, Honolulu, HI 96813, USA; pagano@hawaii.edu; 3Department of Microbiology and Immunology, The State University of New York at Buffalo, Buffalo, NY 14260, USA; runpuche@buffalo.edu (R.C.); yijunsun@buffalo.edu (Y.S.); 4Department of Computer Science and Engineering, The State University of New York at Buffalo, Buffalo, NY 14260, USA; 5Department of Biostatistics, The State University of New York at Buffalo, Buffalo, NY 14260, USA; 6Quantitative Health Sciences, Mayo Clinic, Jacksonville, FL 32224, USA; goodison.steven@mayo.edu; 7Nonagen Bioscience Corporation, Los Angeles, CA 90010, USA

**Keywords:** biomarkers, bladder cancer, multiplex, protein, urinalysis

## Abstract

Background: The Oncuria™ urine test for the detection of bladder cancer measures a multiplex protein signature. In this study, we investigated the influence of urinary cellularity, protein, and hematuria on the performance of the Oncuria™ test in an ex vivo experimental model. Materials and Methods: Pooled urine from healthy subjects was spiked with cultured benign (UROtsa) or malignant cells (T24), cellular proteins, or whole blood. The resulting samples were analyzed using the Oncuria™ test following the manufacturer’s instructions. Results: Urine samples obtained from healthy subjects were negative for bladder cancer by Oncuria™ test criteria. The majority of the manipulated conditions did not result in a false-positive test. The addition of whole blood (high concentration) did result in a false-positive result, but this was abrogated by sample centrifugation prior to analysis. The addition of cellular proteins (high concentration) resulted in a positive Oncuria™ test, and this was unaffected by pre-analysis sample centrifugation. Conclusions: The Oncuria™ multiplex test performed well in the ex vivo experimental model and shows promise for clinical application. The identification of patients who require additional clinical evaluation could reduce the need to subject patients who do not have bladder cancer to frequent, uncomfortable and expensive cystoscopic examinations, thus benefiting both patients and the healthcare system.

## 1. Introduction

Bladder cancer detection and post-treatment surveillance currently require invasive procedures for diagnosis. Non-invasive, urine-based tests are highly desirable for both patients and the healthcare system. Currently, voided urinary cytology (VUC) is the most widely used non-invasive urine test with specificities ranging from 85–100%, however due to its limited test sensitivity (13–75%), it is not used as a stand-alone test [1,2]. Four additional urinary tests have been used in this context: (i) the bladder tumor antigen (BTA) test, which detects urinary complement factor H-related proteins [3,4]; (ii) the nuclear matrix protein 22 (NMP22) test, which detects a cellular nuclear protein [5,6]; (iii) the UroVysion fluorescence in situ hybridization (FISH) assay, which detects aneuploidy in chromosomes 3, 7, and 17, and the loss of the 9p21 locus in urothelial cells [7]; and (iv) the ImmunoCyt/uCyt+ test, which detects the fluorescence of monoclonal antibodies targeting high-molecular-weight form of carcinoembryonic antigen as well as bladder tumor cell-associated mucins [8]. Because of limited test sensitivity, perhaps partially due to benign conditions adversely affecting the test [9,10], the BTA, NMP22, UroVysion FISH, and ImmunoCyt/uCyt+ tests have limited uptake with clinicians, and their utilization have waned over the past decade [11,12]. Bladder EpiCheck was commercially introduced to the market in Europe in 2017. The test analyzes subtle disease-specific changes in DNA methylation markers, allowing for the detection of 92% of high-risk cancers [13]. However, though the sensitivity and specificity were great in the high-risk cohort in the initial cohort, overall sensitivity and specificity in low- and high-grade bladder cancer samples from follow up studies were 62–67% and 82–88%, respectively [14]. In addition, Bladder EpiCheck is not FDA-approved yet. We have previously shown that urinary blood can adversely affect the accuracy of the BTA test and both blood and urine cellularity can affect the accuracy of the NMP test. The reliance of these tests on the measurement of one analyte may make them prone to high false-positive rates, and it is likely that no single-biomarker test would have sufficient predictive power to be applied to the management of individual patients [15,16].

In a phased approach [17,18,19,20,21,22,23,24], we have previously coupled high throughput, discovery-based technology (i.e., genomics and proteomics) with bioinformatics in order to derive diagnostic signatures that show promise for the accurate detection of bladder cancer in voided urine samples. With follow-up studies using retrospective cohorts, we have identified 10 analytes (angiogenin, ANG; apolipoprotein E, APOE; alpha-1 antitrypsin, A1AT; carbonic anhydrase 9, CA9; interleukin 8, IL8; matrix metallopeptidase 9, MMP9; matrix metallopeptidase 10, MMP10; plasminogen activator inhibitor 1, PAI1; syndecan 1, SDC1; and vascular endothelial growth factor, VEGF) in voided urine. After the validation of the 10 urine-based protein biomarkers, we have developed a multiplex, bead-based immunoassay that simultaneously monitors the concentrations of the 10 protein biomarkers in a single voided urine sample. The concentration of each analyte is incorporated into a weighted algorithm to generate a bladder cancer risk score [25,26,27]. To date, we have validated the diagnostic signature in over 2600 individuals [26,27,28], and have completed analytical validation of the test [29].

Here, to test the effects of potential interference from contents of urine, such as shed cells (benign and cancer) and blood (hematuria), we evaluated the performance of the Oncuria™ test in an established ex vivo experimental model [9,10].

## 2. Materials and Methods

### 2.1. Clinical Sampling and Processing

Under IRB approval and with informed consent, whole blood and voided urine samples were collected from three healthy male subjects with a mean age of 39. Microscopic examination and urinary dipstick tests of urine samples revealed no abnormalities. Urine samples were centrifuged for 10 min at 1000× *g*. Pertinent clinical information for each subject was recorded.

### 2.2. Cell Lines and Culture

Human bladder cancer cell line T24 (ATCC, Manassas, VA, USA) was available commercially. The benign human bladder cell line, UROtsa, was a generous gift from Dr. Donald Sens at the University Of North Dakota School of Medicine (Grand Forks, ND, USA). T24 and UROtsa cells were maintained in RPMI1640 media (ATCC, Manassas, VA, USA) and low-glucose DMEM (Thermo Fisher Scientific, Waltham, MA, USA), respectively. All media were supplemented with 10% fetal bovine serum, 100 units/mL of penicillin, and 100 μg/mL of streptomycin. All cells were incubated at 37 °C in a humidified atmosphere of 5% CO_2_ in air.

### 2.3. Experimental Model

Th ex vivo experimental model followed previously reported protocols [9,10]. Briefly, 200 mL of freshly voided urine samples from three healthy controls were collected in sterile containers. The urine samples were pooled, mixed, and distributed into 10 mL aliquots in 15 mL centrifuge tubes. The human bladder cell lines were washed, trypsinized, and counted. For whole cells, 1 × 10^4^ (low concentration), 1 × 10^5^ (medium concentration), and 1 × 10^6^ (high concentration) cells were each added to triplicate 10 mL pooled urine samples. For cell lysate analyses, 1 × 10^6^ cells of each cell line were lysed by RIPA buffer (Pierce, Rockford, IL, USA), and total protein concentration was measured. The total proteins extracted from 1 × 10^6^ cells of UROtsa and T24 cells were 431 μg and 369 μg, respectively, with a mean total protein extract of 306 μg. In subsequent experiments, 3.06 μg, 30.6 μg, and 306 μg of cellular proteins were used, corresponding to 1 × 10^4^ (low), 1 × 10^5^ (medium), and 1 × 10^6^ (high) cells. To monitor hematuria influence, pooled whole blood (5, 20, and 50 µL) from 3 healthy subjects was added to 10 mL urine samples, which equates to what is seen in the clinic, microscopic hematuria, and gross hematuria. The urine samples were spiked on ice and immediately stored in −80 °C until Oncuria™ test. All spiking conditions were performed and analyzed in triplicate and unspiked control was tested in five-replicate. Figure 1 illustrates the experimental model approach.

### 2.4. Multiplex Immunoassay

The concentrations of the 10 proteins (A1AT, APOE, ANG, CA9, IL8, MMP9, MMP10, PAI1, SDC1, and VEGFA) were monitored using an analytically validated multiplex bead-based immunoassay (Oncuria™) from R&D Systems Inc. (Minneapolis, MN, USA) [29,30]. Urine samples were centrifuged for 10 min at 1000× *g*. Urine samples were handled on ice prior to diluting 2-fold with R&D Assay Diluent. Samples, standards, and controls (50 μL) were added to the 96-well plate in duplicate. The multiplex immunoassay was conducted according to the manufacturer’s instructions. A seven-point standard curve across the 4-log dynamic range of the assays was included in the assay design. Plates were read on the Luminex^®^ 200 analyzer (Luminex Corp, Austin, TX, USA). Calibration curves were generated for optimal fit in conjunction with Akaike’s information criteria (AIC) values [31].

### 2.5. Data Analysis

A Student t-test was used to compare analyte expression levels in all study conditions. We previously generated a molecular signature model with each sample represented as a vector with 10 dimensions representing the 10 biomarkers. The cutoff of the model to classify tumor burden was >0.4676 [32]. Each of the study conditions were imported into the model and analyzed. Statistical significance in this study was set at *p* < 0.05, and all reported *p* values were 2-sided. All analyses were performed using SAS software version 9.3 (SAS Institute Inc., Cary, NC, USA).

## 3. Results

The Oncuria™ test was performed on all samples from the ex vivo experimental model (Figure 1). To reduce skewness when comparing these results to previous data, log transformation, log_10_(data+0.01), was applied to data for each biomarker. The average of the absolute concentration of the 10 biomarkers for each of the three conditions was reported (Table 1). The addition of whole blood at 5 μL (low), 20 μL (moderate), and 50 μL (high) resulted in a progressive, and significant increase in the urinary concentrations of MMP9, VEGF, PAI1, ApoE, A1AT, ANG, and MMP10. Pre-analysis centrifugation significantly reduced the increase in spiked urinary sample concentrations of VEGFA, PAI1, and ANG (Table 1). The addition of benign cell lysate resulted in a significant increase of MMP9, VEGFA, ApoE, ANG, and MMP10, and the addition of cancer cell lysate resulted in a significant increase of MMP9, CA9, PAI1, ApoE, A1AT, ANG, and MMP10. Pre-analysis centrifugation did not significantly reduce the levels of the proteins from benign cell or cancer cell lysate spiking (Table 1). The addition of benign whole cells resulted in a significant increase of MMP9, VEGFA, ApoE, ANG, and MMP10, and the addition of cancerous whole cells resulted in the significant increase of MMP9, VEGFA, CA9, SDC1, PAI1, ApoE, ANG, and MMP9. Pre-analysis centrifugation of spiked samples resulted in a reduction in VEGF, but did not reduce the levels of MMP9, ApoE, ANG, or MMP10. Pre-analysis centrifugation of samples spiked with the T24 tumor cell line reduced levels of MMP9, SDC1, PAI1, ApoE, and ANG (Table 1).

Interestingly, IL8 was the only urine sample analyte that was not affected by any of the manipulated conditions. Despite the interference caused to individual analyte levels by the addition of whole blood or benign cell protein (whole cells or lysate), when the concentrations of all 10 analytes were computed by the Oncuria ™ diagnostic algorithm, only a high level of whole blood produced a false positive result (risk score > 46.76), and a simple pre-analysis centrifugation step corrected the call to a true negative test result. As expected, a high concentration of T24 cancer cell line lysate resulted in a positive Oncuria™ test, with or without pre-analysis centrifugation (Table 1).

## 4. Discussion

In this study, we evaluated the performance of the Oncuria™ test in an established, ex vivo experimental model [9,10]. Some of the individual biomarkers included in the Oncuria™ test were significantly elevated by the addition to urine samples of whole blood, cell lysate, or whole cells, but the previously prescribed analysis of the complete, multiplex diagnostic signature was robust to errors. A non-invasive test that can reliably classify patients for bladder cancer risk may reduce the number of patients who are unnecessarily subjected to frequent, uncomfortable, and expensive cystoscopic examinations.

Applying an established ex vivo experimental model [9,10], we evaluated the impact of a number of potentially confounding factors on the performance of the Oncuria™ urine test designed for bladder cancer detection. Although the model does not exactly mimic the actual physiological situation, the aim was to test whether a false-positive Oncuria™ test could originate from the presence of blood or non-cancerous cells of urothelial origin. Previous studies using a similar experimental model showed that a positive NMP22 test could result from the presence of blood or benign cells (whole or lysate) within a urine sample [9,33], and a positive BTA test could result from the presence of blood [10,34]. These two single-analyte urine tests may be susceptible to conditions that can be present in benign conditions, e.g., urinary tract infection, kidney stone. On the other hand, three other tests (UroVysion, ImmunoCyt, and Bladder EpiCheck) can detect genetic and epigenetic alterations in bladder cancer. Because cancers develop due to the ‘accumulation’ of multiple genetic and epigenetic alterations, there is a lot of variance in the performance of these tests in different studies [14].

As part of the phased development of the Oncuria™ test, the diagnostic signature was confirmed not only in voided urine samples [17,18,19,20,21,22,23,24,25,26,27,28,29,30,32] but also in excised bladder tumor tissue in a series of immunohistochemical studies [35,36,37,38,39]. Validation studies and analytical evaluation of multiplex kits and technical platforms led to the development of the multiplex Oncuria™ test (R&D Systems) [29,32]. In this study, we evaluated the established test in an ex vivo experimental model to assess robustness in the presence of factors that have the potential to adversely affect the test. The addition of whole blood, cellular lysate, and whole cells to urine samples did result in concentration variability of individual analytes that comprise the test; however, a simple centrifugation step prior to urine sample analysis eliminated the variability. Accordingly, a brief centrifugation step is recommended in the Oncuria™ test protocol. Regardless, the multiplex nature of the test, combined with the incorporation of data computation using a weighted algorithm was able to protect the test from false-positive calls associated with manipulated benign conditions.

The non-invasive detection of bladder cancer using diagnostic biomarkers still remains a challenge. The inadequate power of single markers may partly explain this. The notion that the presence or absence of one molecular biomarker will aid diagnostic or prognostic evaluation has not proved to be the case. This makes sense when one considers the complex interactions between various molecules within, and across molecular pathways, the redundancy of specific cellular mechanisms, and the oligoclonality of solid tumors. The deployment of multiplex diagnostic signatures, akin to ‘fingerprints’ of cancer, are more likely to achieve superior test sensitivity values and be less prone to errors caused by sample variability.

As bladder cancer is a common neoplastic disease encountered worldwide, the development of an accurate, non-invasive diagnostic test would benefit both patients and healthcare systems. Such a test could be incorporated into current workflows to rule out patients who do not require further evaluation, reducing the need to subject patients who do not have bladder cancer to frequent, uncomfortable, and expensive cystoscopic examinations. The multiplex Oncuria™ test shows promise for application in the urological setting. Additional studies are underway to evaluate the potential added value of the test in current clinical decision making.

## Figures and Tables

**Figure 1 diagnostics-11-01023-f001:**
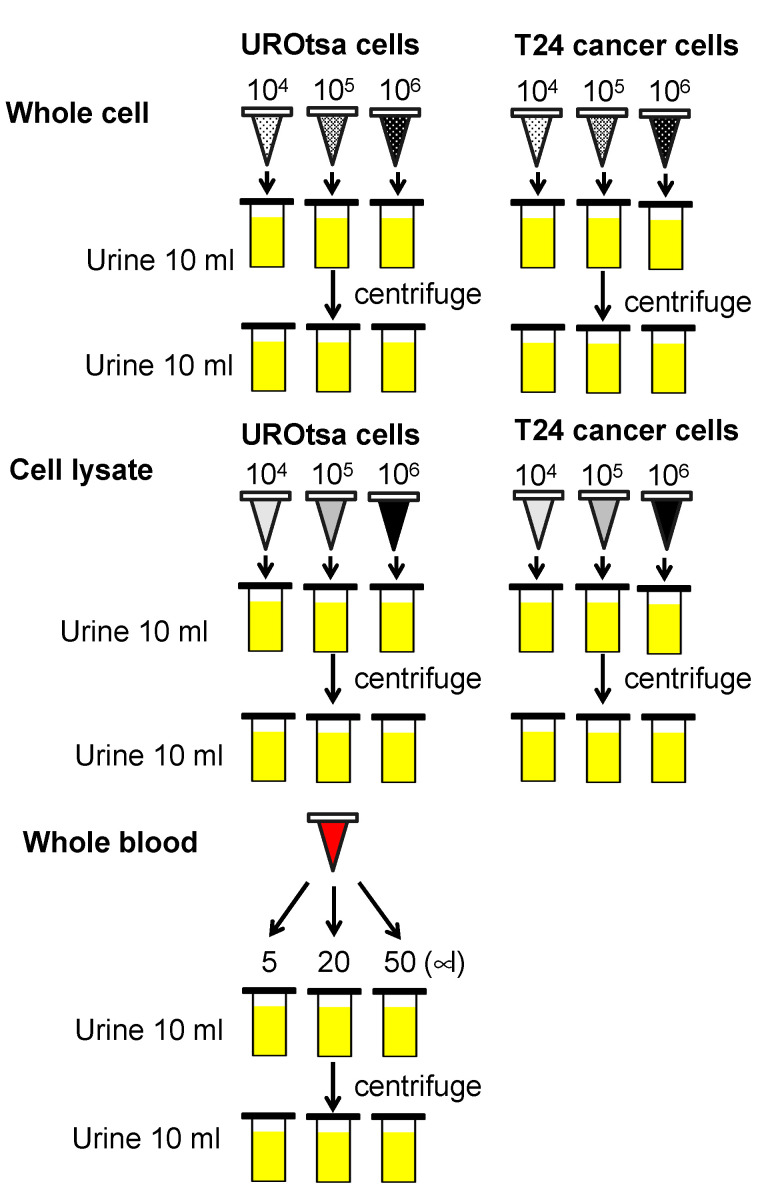
**Schematic of experimental model.** Low concentration (10^4^), medium concentration (10^5^) and high concentrations (10^6^) of UROtsa benign human bladder cells, or low concentration (10^4^), medium concentration (10^5^) and high concentrations (10^6^) of T24 human bladder cancer line were added to 10 mL of pooled urine from three healthy subjects. The cellular lysate associated with low (3.06 μg), medium (30.6 μg) and high (306 μg) concentrations of UROtsa or T24 cell lines were added to 10 mL of pooled urine from three healthy subjects. Whole blood (5, 20 and 50 µL) were added to 10 mL of pooled urine from healthy subjects.

**Table 1 diagnostics-11-01023-t001:** Mean urinary concentrations of 10 biomarkers assessed by Oncuria™.

Condition	Spike	Centrifuge	Level	Category	MMP9	CXCL8_IL8	VEGFA	IX_CA9	SDC1	PAI1	APOE	A1AT	ANG	MMP10	Risk Scorre
1	Pooled Urine				16.7	5.62	38.2	0.629	7818	6.83	516.4	13,139	49.5	3.8	12.8
2	Blood	No	Low		26.2 ^	6.07	36.9	0.659	7704	6.49	801.7	25,052	43.1	9.1	18.8
3	Blood	No	Med		42.1 ^	5.82	38.9	0.523	7907	7.96	1449.7 ^	52,793	62.0	10.8	28.6
4	Blood	No	High		77.8 ^	5.75	47.9 #	0.546	8556	13.5 ^	3610.1 ^	215,761.2 #	139.6 ^	11.9 *	**51.7**
5	Blood	Yes	Low		25.2 ^	5.49	31.4 ~	0.318	7991	4.81	725.3	25,554	28.5	7.0	10.8
6	Blood	Yes	Med		43.1 ^	5.58	32.4 *	0.477	7560	4.70	1083.6 ~	58,279	26.5 *	13.9 *	17.5
7	Blood	Yes	High		69.9 ^	5.57	33.5	0.318	8324	6.21	2387.7 ^	127,200.2 *	34.0	10.0	28
8	Cell Lysate	No	Low	Benign	20.3 *	5.84	44.7 *	1.023	7218	7.56	892.2 ~	15,936	88.5 ~	17.6 ~	29.6
9	Cell Lysate	No	Med	Benign	18.9	5.56	40.3	0.432	6972	7.72	904.6 ~	15,544	110.7 ^	86.6 ^	27.4
10	Cell Lysate	No	High	Benign	16.4	5.65	42.6	1.023	7350	8.06	958.9 ~	16,179	104.9 #	469.9 ^	37.6
14	Cell Lysate	Yes	Low	Benign	19.4 *	5.72	43.6 *	0.864	7503	8.31	870.6 ~	16,263.8 *	122.8 ^	25.4 ~	29
15	Cell Lysate	Yes	Med	Benign	19.9 *	5.74	44.8 *	0.796	7138	7.84	839.3 ~	16,500.6 *	123.5 ^	123.4 ^	30.3
16	Cell Lysate	Yes	High	Benign	18.4	5.73	42.9	0.455	7606	8.68	931.7 ~	162,96.6 *	131.6 ^	1169.0 ^	31.7
11	Cell Lysate	No	Low	Cancer	20.3 *	5.39	40.9	0.568	7837	8.14	887.1 ~	16,845.5*	88.5 ~	10.9	23.4
12	Cell Lysate	No	Med	Cancer	20.6 ~	5.49	41.3	1.9 *	7566	8.45	921.5 ~	15,523	95.6 #	15.5 ~	37
13	Cell Lysate	No	High	Cancer	19.9 *	5.94	43.2	13.9 ^	7098	10.8 ~	864.9 ~	14,751	92.4 #	15.3 ~	**61.4**
17	Cell Lysate	Yes	Low	Cancer	21.5 #	5.23	63.3 ^	1.652	10,358.7 *	7.70	690.1	15,801	107.9 #	33.6 #	19.4
18	Cell Lysate	Yes	Med	Cancer	20.0 ~	5.49	40.4	2.5 ~	7504	8.35	852.1 ~	15,237	96.4 ~	11.4	37.8
19	Cell Lysate	Yes	High	Cancer	18.0	5.53	41.1	11.9^	7488	11.1 #	824.0 *	15,701	84.5 *	14.8	**56.8**
20	Whole Cell	No	Low	Benign	19.1 *	5.38	41.9	0.546	7119	7.49	828.0 ~	14,936	91.9 ~	20.7 *	24
21	Whole Cell	No	Med	Benign	20.0 ~	5.49	41.7	1.251	7436	7.87	927.5 ~	17,132	101.5 #	56.5 ^	34.5
22	Whole Cell	No	High	Benign	19.4 *	5.81	46.4 ~	0.591	7429	8.58	983.9 #	16,529	122.8 ^	666.9 ^	34.1
26	Whole Cell	Yes	Low	Benign	18.2	5.58	42.1	0.364	7832	8.17	808.4 ~	16,903. 8 *	116.6 ^	16.6	19.2
27	Whole Cell	Yes	Med	Benign	20.0 ~	5.53	43.8 *	0.637	7129	7.71	820.1 ~	15,723	112.5 ^	70.5 ^	27.1
28	Whole Cell	Yes	High	Benign	19.6 *	5.64	43.2	1.387	7392	8.08	829.4 ~	15,956	123.7 ^	837.6 ^	38.6
23	Whole Cell	No	Low	Cancer	24.0 ^	5.79	71.7 ^	1.538	11,688.9 #	9.0 *	796.8 *	16,460	137.2 ^	23.3 *	19.9
24	Whole Cell	No	Med	Cancer	25.1 ^	6.17	71.9 ^	2.922	11,278.3 ~	8.89	790.2 *	16,320	115.7 ^	24.5 ~	24.9
25	Whole Cell	No	High	Cancer	18.3	5.12	56.7 ^	5.9 ^	10,854.3 ~	10.8 #	734.7	12,662	122.9 ^	24.6 ~	35.4
29	Whole Cell	Yes	Low	Cancer	21.2 #	5.81	65.5 ^	2.238	9888	6.09	551.5	15,600	70.0	28.2~	15.5
30	Whole Cell	Yes	Med	Cancer	17.5	4.80	51.8 ^	1.954	9533	5.25	519.2	14,850	47.6	26.3 *	13.9
31	Whole Cell	Yes	High	Cancer	18.2	5.39	49.9 #	6.6 ^	9589	4.74	459.4	14,629	32.9	28.9 ~	17.9

Bold, positive Oncuria test; *, *p* < 0.05; ~, *p* < 0.01; #, *p* < 0.001; ^, *p* < 0.0001.

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
