# Peer review of "Influencing Factors on the Oncuria™ Urinalysis Assay: An Experimental Model"

_diagnostics, 2021, doi:10.3390/diagnostics11061023_

Round 1
Reviewer 1 Report
In this paper, the authors propose a methodology to support previous published data regarding a multiplex protein signature test for detection of bladder cancer in urine samples.
Overall, the approach the authors propose is interesting, although some clarifications are needed:
- In the Introduction section the authors talk about two additional urinary tests for BlCa detection: the NTA and the NMP22 tests. What about the other FDA approved available urinary tests? (e.g. Urovysion and ImmunoCyt).
- Very few samples were used in this study (n = 3). Is it possible to improve the number of healthy donors samples in order to have a more robust evaluation and statistical analysis?
- The authors do not mention the incubation times and temperatures of the urine samples with the whole cells, cell lysates and whole blood. This is an important information because it should influence the following experiments and results.
- In the Discussion section, the authors should discuss more about the previous papers regarding this biomarker panel, and the other panels which have been used, especially the ones approved by FDA, or the CE IVD marked tests. Was this panel tested in an at-risk population of non-BlCa patients (e.g. patients with hematuria, bladder inflammation, etc)? The Discussion needs an improvement.
Author Response
First of all, I would like to thank you for the time and effort you spent to evaluate our manuscript. We found the vast majority of comments/suggestions/questions very relevant, and we have clarified in the revised version of the manuscript the most serious points raised by you. We have now thoroughly revised and added more details in the introduction, methods, results and discussion. We hope that the revised manuscript is now suitable for publication in Diagnostics.
Response to Reviewer 1:
1. In the Introduction section the authors talk about two additional urinary tests for BlCa detection: the NTA and the NMP22 tests. What about the other FDA approved available urinary tests? (e.g. Urovysion and ImmunoCyt).
Response: We have added the information about Urovysion and ImmunoCyt in the Introduction.
2. Very few samples were used in this study (n = 3). Is it possible to improve the number of healthy donors samples in order to have a more robust evaluation and statistical analysis?
Response: Thank you very much for your comments and sorry for the confusion. The whole point of this manuscript is the ex vivo model. In this study, to evaluate the effects of potential interference from contents of urine, such as shed cells (benign and cancer) and blood (hematuria), we took urine from normal subjects, POOLED them together and spiked with benign and cancer and blood. There are 5-replicates in control urine (unspiked) and triplicates in spiked urine, which are based on the standard protocol for interference testing. These descriptions are added to the manuscript.
3. The authors do not mention the incubation times and temperatures of the urine samples with the whole cells, cell lysates and whole blood. This is an important information because it should influence the following experiments and results.
Response: The urine samples were spiked on ice and immediately stored in -80°C. Therefore, we expect that we simply added the components of cells or blood, and do not expect that the addition of cells or blood altered the original components of urine. We modified the descriptions in methods to avoid the confusion.
4. In the Discussion section, the authors should discuss more about the previous papers regarding this biomarker panel, and the other panels which have been used, especially the ones approved by FDA, or the CE IVD marked tests. Was this panel tested in an at-risk population of non-BlCa patients (e.g. patients with hematuria, bladder inflammation, etc)? The Discussion needs an improvement.
Response: Again, the objective of this manuscript is to evaluate the effects of potential interference from contents of urine on the Oncuria™ test. We have already performed analytical and clinical validation of the Oncuria™ test, which have been published in other manuscripts (Furuya et al., Pract Lab Med, 2020, PMCID: PMC7691749, and Hirasawa et al., J Transl Med, 2021, PMCID: PMC8025333). I added the references in the discussion.
Reviewer 2 Report
Dear authors, I read carefully your paper and I found interesting. In reality I am a clinician and I am not qualified enough to judge all the aspects, anyhow I can offer you some improvements for the paper in general and from the clinical point of view.
Introduction: maybe you can mention, between non-invasive tests, even EpiCheck. It is an important test and is between the most reliable.
Please fix last paragraph that should be only the aim of the study without comments on the test. Finally, I believe that you can spend some more words on the 10 analytes included in the test.
Materials and Methods: when talking about “clinical information” of the patients they should be mentioned.
Discussion: I believe that this section could be implemented. You should discuss more on other test results and compare with Oncuria. Don’t forget, even here, to include EpiCheck. Finally, you should include a small section mentioning main limitations of your study.
Author Response
First of all, I would like to thank you for the time and effort you spent to evaluate our manuscript. We found the vast majority of comments/suggestions/questions very relevant, and we have clarified in the revised version of the manuscript the most serious points raised by you. We have now thoroughly revised and added more details in the introduction, methods, results and discussion. We hope that the revised manuscript is now suitable for publication in Diagnostics.
Response to Reviewer 2:
- Introduction: maybe you can mention, between non-invasive tests, even EpiCheck. It is an important test and is between the most reliable.
Response: We added the information about UroVysion and ImmunoCyt as well as bladder EpiCheck. But please note that EpiCheck was evaluated with over 1,300 subjects (Oncuria™ test was evaluated with 2,600 subjects) with lower sensitivity and specificity in the follow up studies, and is not FDA-approved yet.
- Please fix last paragraph that should be only the aim of the study without comments on the test. Finally, I believe that you can spend some more words on the 10 analytes included in the test.
Response: We have moved the comments on the test from introduction to discussion. We also have added the additional description on the 10 analytes.
- Materials and Methods: when talking about “clinical information” of the patients they should be mentioned.
Response: We added the mean age and number of males in the text.
- Discussion: I believe that this section could be implemented. You should discuss more on other test results and compare with Oncuria. Don’t forget, even here, to include EpiCheck. Finally, you should include a small section mentioning main limitations of your study.
Response: We have already discussed about BTA and NMP22 kits, which are urine protein-based diagnostics of bladder cancer. The objective of this manuscript is to evaluate the effects of potential interference from contents of urine on the Oncuria™ test. Previous studies showed that a false-positive NMP22 test could result from the presence of blood or benign cells (whole or lysate) within a urine sample, and a false-positive BTA test could result from the presence of blood. On the other hand, the addition of whole blood, cellular lysate and whole cells to urine samples did result in concentration variability of individual analytes that comprise the test, however, a simple centrifugation step prior to urine sample analysis eliminated the variability. UroVysion, ImmunoCyt, and Bladder EpiCheck detect genetic and epigenetic alterations in bladder cancer. Because cancers develop due to the “accumulation” of multiple genetic and epigenetic alterations, there is a lot of variances in the performance of these tests in different studies. The discussion above was added to the discussion section in the manuscript.
Round 2
Reviewer 1 Report
The authors have responded to all the questions and suggestions.
Reviewer 2 Report
No further requirements